

# Up-regulated IL-17 and Tnf signaling in bone marrow cells of young male osteogenesis imperfecta mice

Chenyi Shao[1],[*], Yi Liu[1],[*], Jiaci Li[2], Ziyun Liu[1], Yuxia Zhao[1], Yaqing Jing[1], Zhe Lv[1], Ting Fu[1], Zihan Wang[1] and Guang Li[1]

[1] Tianjin Medical University, Tianjin, China
[2] Tianjin Pediatric Research Institute, Tianjin Children's Hospital, Tianjin, Longyan Road, Beichen District, Tianjin, China
[*] These authors contributed equally to this work.

Corresponding author
Guang Li, lig@tmu.edu.cn

## ABSTRACT

Osteogenesis imperfecta (OI) is a congenital bone dysplasia mainly caused by either defective production or assembly of type I collagen. The skeletal phenotypes especially fractures are often seen in OI adolescents. Studies have found that an increased number of osteoclasts and excessive bone resorption existed in collagen-related OI, which has not been well understood. Emerging evidence has suggested that inflammation may be associated with OI. We speculated that the bone marrow (BM) niche had similar inflammatory changes and performed RNA-sequencing (RNA-seq) in BM cells derived from young male mice to analyze the related differentially expressed genes (DEGs) and pathways. Data showed that there were 117 shared DEGs ($Q \le 0.05$, $|\log_2\mathrm{FC}| \ge 1$) in BM cells isolated from two types of OI murine models that respectively simulate different OI types. Gene Ontology (GO) ($Q \le 0.05$) analysis, Kyoto Encyclopedia of Genes and Genomes (KEGG) ($Q \le 0.05$) analysis and real-time PCR validation indicated the dysregulated biology process of cellular response to interferon (Ifn) together with upregulated IL-17 signaling, tumor necrosis factor (Tnf) signaling and osteoclast differentiation in OI BM niche. Either defective collagen production or abnormal collagen assembly shared similar alterations in gene profiles and pathways involving inflammation and osteoclast activation. Data presented here not only contributed to understanding of the mechanism of the enhanced bone absorption in the bones of OI, but also provided more evidence to develop potential anti-inflammation therapies.

## HIGHLIGHT

1. There were 117 shared differentially expressed genes in bone marrow (BM) cells isolated from two types of OI young male murine models.
2. The upregulated IL-17 signaling, Tnf signaling and osteoclast differentiation were significantly enriched in OI BM cells.

3. These dysregulated DEGs and pathways in BM cells might be associated with the excessive bone resorption of the OI mice.

## INTRODUCTION

Osteogenesis imperfecta (OI) is a congenital disorder characterized by bone fragility (*Forlino et al., 2011*). The patients generally suffer from recurrent bone fractures and deformities. Up to 21 genes have been associated with OI, but most of the total cases are caused by heterozygous mutation in either of the genes coding for the type I collagen alpha chains, *COL1A1* or *COL1A2* (*Forlino et al., 2011*). The pathogenic variants of the two genes often cause either insufficient synthesis or abnormal structure of skeletal collagen, which is correlated with mild OI type I and more severe type II–IV respectively (*Saito & Marumo, 2015*; *Nijhuis et al., 2019*).

Multiple reports have highlighted the impaired bone formation and excessive bone absorption in defective collagen-related OI (*Takeyari et al., 2021*; *Iwamoto, Takeda & Ichimura, 2002*), which are the main targets in OI treatments. Type I collagen fibrils are produced by bone-forming osteoblasts, making the dysfunctional osteoblasts and their weakened mineralization the main focus of OI etiological research. Conversely, the mechanism of enhanced bone destruction in OI bones has not been well understood, although anti-resorptive drugs have been widely applied in clinical interventions.

Some recent studies have indicated the contribution of inflammation and inflammatory factors in OI bone phenotype. Increased transformation growth factor-beta (TGF β) and excessive TGF signaling have been regarded as a promising target to enhance bone formation in some OI models and patients (*Grafe et al., 2014*; *Marom, Rabenhorst & Morello, 2020*). Also, the elevated serum level of interferon (IFN) and tumor necrosis factor (TNFα) have been found in patients and mouse models separately, suggesting the chronic inflammation state in OI (*Zhytnik et al., 2020*). Inflammation has been closely linked to bone biochemical changes and bone loss in rheumatoid arthritis (RA) and osteoporosis (*Weyand & Goronzy, 2021*; *Tilg et al., 2008*). The progenitor cells of osteoblasts and osteoclasts derived from hematopoietic stem cell (HSC) lineage are both situated in the same bone marrow (BM) niche (*Ono & Nakashima, 2018*). The inflammatory alterations of the BM niche that have not been well explored may directly or indirectly act on the bone manifestations of OI.

In the process of bone marrow development, the transformation of red bone marrow to yellow bone marrow is most obvious in infancy and childhood (*Berg, 2021*). Compared with the old bone marrow, the level of ROS in the young bone marrow is lower, and the cells are less affected by aging, which can better reflect the initial bone marrow state under genetic defects (*Yao et al., 2021*). Also, the skeletal symptoms of OI are most obvious in childhood and can be relieved in adulthood (*Paterson, McAllion & Stellman, 1984*). Studying the bone marrow of young mice will help to understand the cause and process of OI. Here, we performed RNA-sequencing (RNA-seq) and differential gene expression analyses in femoral BM cells isolated from young mice of two types of OI mouse models and wild-type mice to explore the changes in the BM niche. Both the $Col1a1^{+/-365}$ mice

and heterozygous oim mice ($Col1a2^{oim/+}$) display osteogenesis deficiency accompanied by an increased number of osteoclasts. The differentially expressed genes (DEGs) shared by the two OI models were involved in IL-17 signaling pathway, Tnf signaling pathway and osteoclast differentiation. The DEGs related to the three pathways were all upregulated in OI BM cells when compared with normal cells, suggesting the activation of these signalings. These dysregulated genes and pathways in mutant BM cells are likely to play an important role in the pathological changes of OI bones.

## MATERIALS AND METHODS

### OI mice model

Adult wild-type (wt) C57BL/6 mice were purchased from the Laboratory Animal Center of the Academy of Military Medical Science (China). The B6C3Fe a/a-$Col1a2^{oim}$/J mice (#001815) were bought from the Jackson Laboratory (Bar Harbor, ME, USA) and maintained on a congenic C57BL/6 background. The $Col1a2^{oim/+}$ mice were heterozygotes carrying a single mutant $Col1a2$ allele and performed a mild form of OI. The $Col1a1^{+/-365}$ mice with a $Col1a1$ gene knock-down can simulate OI type I. Heterozygous mice were bred by crossing heterozygous individuals and genotyped as previously described (*Chipman et al., 1993*; *Liu et al., 2019*). All mice were housed in specific pathogen-free conditions and sacrificed by $CO_2$ in 4 weeks old to sample. Mice were fed with full-price diet and autoclaved water. The number of mice per cage did not exceed five. All experiments were performed following the approval of Animal Care and Use Committee of Tianjin Medical University (TMUaMEC 2017012). Only male mice were used in the present study.

### Cell isolation

The femurs of 4-weeks and 12-weeks old male mice were harvested and the bone marrow (BM) cells were flushed out with a syringe. The wt and OI modeled mice-derived BM cells were marked as BM^wt, BM^{oim/+} and BM^{+/-365} respectively.

### RNA extraction and RNA sequencing (RNA-seq)

Total RNA of freshly isolated BM cells was extracted using Trizol reagents (Invitrogen, Waltham, MA, USA). Each type of cells isolated from one mouse was seen as one sample and each group contained three samples. RNA samples ($n = 3$/genotype from young animals) with RNA integrity number $\geq 7.0$ and 28S/18S ratios $\geq 1.5$ were sequenced. Another 18 RNA samples ($n = 6$/genotype) were used in quantitative PCR verification.

   A total of 18 cDNA libraries were constructed and separately sequenced by Beijing Genomics Institute (BGI, Beijing, China) using the BGISEQ-500 platform. RNASeqPower Software were used to calculated the statistical power of this experimental design. Sequence data (~75 million reads) were checked for sequencing quantity by FASTQC. Reads with low quality (unknown nucleotides > 10%, or Q20 < 20%), adapter contamination and high N content of unknown bases ($N > 5\%$) were excluded to gain clean reads. The clean reads were then mapped to the mouse reference genome (Mus_musculus, NCBI, GCF_000001635.26_GRCm38.p6) and analyzed by HISAT2 software. Read counts were normalized to TPM (transcripts per kilobase of exon model per million mapped reads).

**Table 1 Sequences of primers used for RT-PCR.**

| Gene | Forward (5'-3') | Reverse (5'-3') | Product length (bp) |
|---|---|---|---|
| Lif | TCAACTGGCACAGCTCAATGGC | GGAAGTCTGTCATGTTAGGCGC | 119 |
| Jun | CCTTCTACGACGATGCCCTC | GGTTCAAGGTCATGCTCTGTTT | 102 |
| Fosb | TTTTCCCGGAGACTACGACTC | GTGATTGCGGTGACCGTTG | 174 |
| Stat1 | TCACAGTGGTTCGAGCTTCAG | GCAAACGAGACATCATAGGCA | 155 |
| Cxcl10 | CCAAGTGCTGCCGTCATTTTC | GGCTCGCAGGGATGATTTCAA | 157 |
| Ccl2 | TTAAAAACCTGGATCGGAACCAA | GCATTAGCTTCAGATTTACGGGT | 121 |
| Ifng | ATGAACGCTACACACTGCATC | CCATCCTTTTGCCAGTTCCTC | 182 |
| Ifit1 | GCCTATCGCCAAGATTTAGATGA | TTCTGGATTTAACCGGACAGC | 75 |
| Irf7 | GAGACTGGCTATTGGGGGAG | GACCGAAATGCTTCCAGGG | 102 |
| Gapdh | CATCACTGCCACCCAGAAGACTG | ATGCCAGTGAGCTTCCCGTTCAG | 153 |

Note:
Lif, leukemia inhibitory factor; Jun, jun proto-oncogene; Fosb, FBJ osteosarcoma oncogene B; Stat1, signal transducer and activator of transcription 1; Cxcl10, chemokine (C-X-C motif) ligand 10; Ccl2, chemokine (C-C motif) ligand 2; Ifng, interferon gamma; Ifit1, interferon-induced protein with tetratricopeptide repeats 1; Irf7, interferon regulatory factor 7; Gapdh, glyceraldehyde-3-phosphate dehydrogenase.

$Q$-value was obtained by false discovery rate (FDR) correction of the $P$-value. Differentially expressed genes (DEGs) ($Q \leq 0.05$, $|\log_2 FC| \geq 1$) were analyzed by DEseq2 software. Gene Ontology (GO) ($Q \leq 0.05$) and Kyoto Encyclopedia of Genes and Genomes (KEGG) ($Q \leq 0.05$) analyses were performed in DEGs shared by the two OI models derived BM cells using BGI Dr. Tom multi-omics data mining system. Gene Set Enrichment Analysis (GSEA) based on the KEGG database was also performed using BGI Dr. Tom and results with $P \leq 0.05$ and $Q \leq 0.25$ were considered statistically significant.

## Quantitative real-time PCR (RT-PCR)

The RNA samples from both young and adult mice were reverse transcribed to cDNA using GoScript reverse transcriptase (Promega, Madison, WI, USA) according to the manufacturer's protocol. RT-PCR analysis was performed using the AceQ RT-PCR SYBR Green Master Mix kit (Vazyme, Nanjing, China). The cycling program referenced our previous report (*Liu et al., 2020*). All samples were evaluated in triplicate and normalized to mouse *glyceraldehyde-3-phosphate dehydrogenase* (*Gapdh)*. All the primers were synthesized from Sangon Biotech Co., Ltd. (China), and the sequences were listed in Table 1.

## Statistical analysis

Statistical analysis was conducted using SPSS version 17.0 software (IBM SPSS Statistics, Chicago, IL, USA). All data were presented as mean ± standard deviation (SD). Two groups were generally compared by unpaired. The D'Agostino-Pearson omnibus normality test was used to test the normal distribution of data. If the data conformed to the normal distribution ($P > 0.1$), the unpaired t-test was used. In the unpaired t-test, the F-test was used to analyze the homogeneity of variance for pairwise comparison. If the variance was homogenous ($P > 0.05$), the unpaired t-test was performed. If variances were not homogeneous ($P < 0.05$), Welch's correction was used to correct them. If the data were not
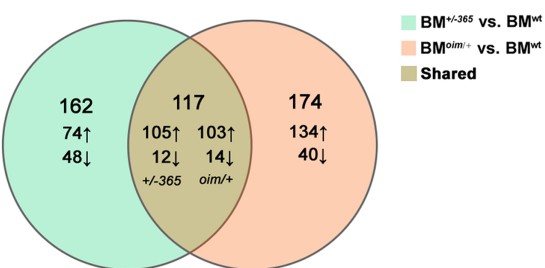

**Figure 1** **The results of RNA-seq of bone marrow (BM) cells.** The Venn diagram of differentially expressed genes (DEGs) in BM cells isolated from 4-weeks old wt mice, heterozygous *Col1a1*$^{+/-365}$ and *Col1a2*$^{oim/+}$ mice ($Q \leq 0.05$, $|\log_2 FC| \geq 1$). There were 117 DEGs shared by the two types of OI mice.

subject to a normal distribution, the Mann-Whitney test was used. $P < 0.05$ was considered statistically significant.

## RESULTS

### Differential gene expression analysis

RNA-seq was performed in BM cells of the three genotypes of mice. The statistical power (depth = 100, cv = 0.1, effect = 2, α = 0.05, power = 0.9) of this experimental design, calculated in RNASeqPower was 0.0875. Compared with BM$^{wt}$, a total of 279 DEGs were identified in BM$^{+/-365}$ and 219 of which were upregulated and 60 were downregulated (Fig. 1). A total of 291 DEGs were differentially expressed in BM$^{oim/+}$ when compared with BM$^{wt}$, 237 of which were upregulated and 54 were downregulated (Fig. 1). There were 117 DEGs shared by the two types of OI mice (Fig. 1). A total of 102 of the common DEGs were concordantly upregulated and 12 were concordantly downregulated, suggesting their consistency of alteration (Data not shown).

The top 20 upregulated shared DEGs in BM$^{+/-365}$ or BM$^{oim/+}$ were listed in Tables 2 and 3 respectively. A total of 17 of them were the same genes that contained several Ifn signaling-related genes (*e.g.*, *Lfit1*, *Lfit3*, *Lfi44* and *Irf7*) (Tables 2 and 3). Notably, the transcription of *Coch*, most abundantly expressed in the inner ear, was obviously increased in the two types of BM cells (Tables 2 and 3). The abnormal *Coch* expression might be associated with OI hearing loss.

### Gene Ontology biology process analysis

Gene Ontology (GE) biology process analysis indicated the enrichment of multiple immune-related genes in such as cellular response to interferon-beta (Ifnβ), immune system process, and immune response, and cellular response to interferon-alpha (Ifnα) in defective BM cells (Fig. 2A). We tested the level of *Lfit1* and *Irf7*, and data showed that both of the two genes were significantly upregulated in mutant BM cells (Figs. 2B and 2C), suggesting the activated Ifn signaling in OI BM.

### KEGG pathway enrichment and GSEA analysis

The significantly enriched pathways by KEGG analysis of the 117 shared DEGs included the IL-17 signaling pathway, Tnf signaling pathway and osteoclast differentiation in OI

**Table 2 The top 20 upregulated shared DEGs in BM$^{+/-365}$ when compared with BM$^{wt}$.**

| Gene | Description | Log2 (Fold chang) | Q-value |
|---|---|---|---|
| *Fgf23* | fibroblast growth factor 23 | 5.83 | 0.0002360794650486 |
| *Ptgds* | prostaglandin D2 synthase (brain) | 5.54 | 0.0021177353272204 |
| *Oas1g* | 2'–5' oligoadenylate synthetase 1G | 4.9 | 0.0009431000835065 |
| *Apol9b* | apolipoprotein L 9b | 4.6 | 0.0002133173066225 |
| *Ifit1* | interferon-induced protein with tetratricopeptide repeats 1 | 4.28 | 0.0003668837536278 |
| *Ifit3* | interferon-induced protein with tetratricopeptide repeats 1 | 4.25 | 0.0015924514011703 |
| *Ifi44* | interferon-induced protein 44 | 4.17 | 0.003021888072243 |
| *Coch* | coagulation factor C homology/cochlin | 4.11 | 0.0260410468387774 |
| *Isg15* | ISG15 ubiquitin-like modifier | 3.98 | 0.0000106275044397477 |
| *Ifit3b* | interferon-induced protein with tetratricopeptide repeats 3b | 3.91 | 0.001908267305679 |
| *Rsad2* | radical S-adenosyl methionine domain containing 2 | 3.38 | 1.62699675914253e−13 |
| *Usp18* | ubiquitin specific peptidase 18 | 3.24 | 0.0010491995647924 |
| *Oas1a* | 2'–5' oligoadenylate synthetase 1A | 3.05 | 0.0016396718614034 |
| *Oasl2* | 2'–5' oligoadenylate synthetase 2 | 3 | 0.0031321485213441 |
| *Gbp6* | guanylate binding protein 6 | 3 | 0.0010429030626467 |
| *Irf7* | interferon regulatory factor 7 | 2.98 | 0.002419157920696 |
| *Cfb* | complement factor B | 2.97 | 0.0133696378926878 |
| *Gbp10* | guanylate-binding protein 10 | 2.97 | 0.0013662770771023 |
| *Zbp1* | Z-DNA binding protein 1 | 2.9 | 0.0088547659475085 |
| *Rtp4* | receptor transporter protein 4 | 2.85 | 0.0013424962435546 |

BM cells (Fig. 3A). There were 11 enriched DEGs involved in the aforementioned pathways (Table 4). GSEA analysis demonstrated that all of the three pathways were markedly upregulated in OI-derived BM cells (Figs. 3B–3D). Besides, the sequencing data demonstrated that both *Il17a* and *Tnf* increased more than two-fold in BM$^{oim/+}$ (Table 5). The *Tnf* expression also got significantly elevated in *Col1a1*$^{+/365}$ mice-derived BM cells (Table 5). In addition, the mRNA level of *Rankl* (*Tnfsf11*) rose in defective BM cells (Table 5). It has been reported that upregulated IL-17 and Tnf signalings contribute to the excessive bone resorption in multiple inflammatory bone loss diseases (*Tsukasaki & Takayanagi, 2019*; *Weitzmann, 2017*), suggesting the potential role of overactivated IL-17 and Tnf pathways in an increased number of osteoclasts under the OI background.

**The changes of DEGs in BM cells isolated from OI modeled mice**

The heatmap as shown in Fig. 4A displayed the differential expression of the 11 DEGs mentioned above. Many of them at least participate in two of the IL-17 signaling pathway, Tnf signaling pathway, and osteoclast differentiation, suggesting their complex crosstalk. RT-PCR validation indicated that *Ccl2* (Fig. 4B), *Jun* (Fig. 4C) *Cxcl10* (Fig. 4D), *Lif* (Fig. 4E) and *Stat1* (Fig. 4F) were apparently upregulated in mutant BM cells, which was consistent with the sequencing data. And *Fosb* (Fig. 4G) and *Ifng* (Fig. 4H) expression also showed an increasing trend in OI mice-derived BM cells. These results preliminarily identified the dysregulated DEGs, which might play important roles in OI bone erosion alterations.

**Table 3 The top 20 upregulated shared DEGs in BM$^{oim/+}$ when compared with BM$^{wt}$.**

| Gene | Description | Log2 (Fold chang) | Q-value |
|---|---|---|---|
| *Ptgds* | prostaglandin D2 synthase (brain) | 7.5 | 6.17204589560265e−9 |
| *Coch* | coagulation factor C homology/cochlin | 6.02 | 2.94335300566472e−8 |
| *Oas1g* | 2'–5' oligoadenylate synthetase 1G | 5.18 | 9.681071247414061e−14 |
| *Fgf23* | fibroblast growth factor 23 | 4.99 | 0.000606927029448 |
| *Ifi44* | interferon-induced protein 44 | 4.46 | 1.23002041345624e−21 |
| *Ifit3* | interferon-induced protein with tetratricopeptide repeats 3 | 4.44 | 1.08339825790861e−9 |
| *Ifit1* | interferon-induced protein with tetratricopeptide repeats 1 | 4.22 | 5.155872680064089e−11 |
| *Ifit3b* | interferon-induced protein with tetratricopeptide repeats 3b | 4.2 | 9.503797111990189e−11 |
| *Gbp6* | guanylate binding protein 6 | 3.63 | 1.1575697141787101e−74 |
| *Apol9b* | apolipoprotein L 9b | 3.52 | 0.0012727812836212 |
| *Cfb* | complement factor B | 3.47 | 9.1080396056311e−39 |
| *Isg15* | ISG15 ubiquitin-like modifier | 3.37 | 7.51494012126258e−18 |
| *Tgtp1* | T cell specific GTPase 1 | 3.36 | 1.28520549551154e−10 |
| *Oasl2* | 2'–5' oligoadenylate synthetase 2 | 3.19 | 1.83115647109809e−26 |
| *Gbp10* | guanylate-binding protein 10 | 3.19 | 3.41807441735383e−13 |
| *Oas1a* | 2'–5' oligoadenylate synthetase 1A | 3.15 | 3.8026707877775104e−15 |
| *Usp18* | ubiquitin specific peptidase 18 | 3.14 | 1.29677254526075e−14 |
| *Iigp1* | interferon inducible GTPase 1 | 3.14 | 0.000013763932948896 |
| *Irf7* | interferon regulatory factor 7 | 3.12 | 6.5309598268511e−36 |
| *Ly6i* | lymphocyte antigen 6 complex, locus I | 3.1 | 3.91587391622785e−21 |

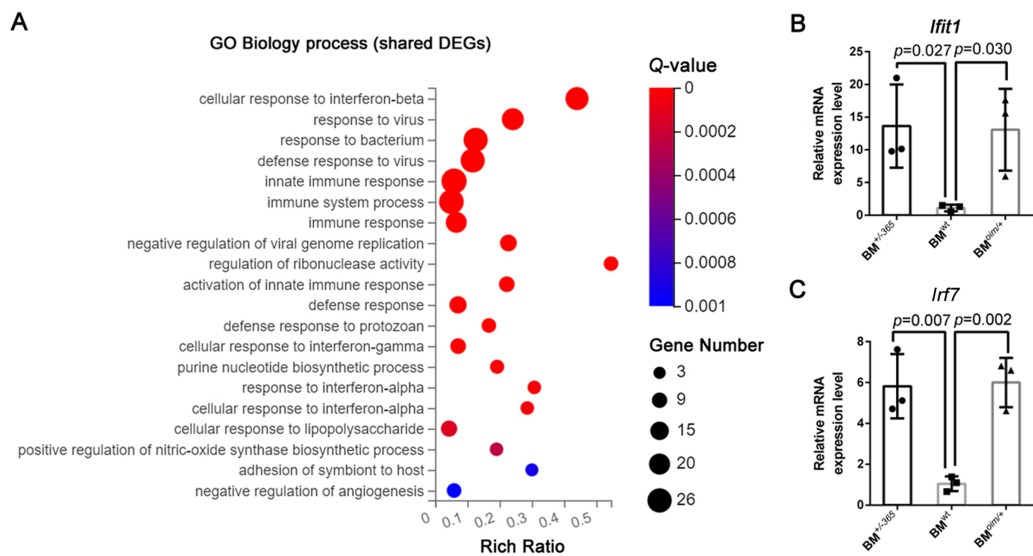

**Figure 2 The results of GO Biology process analysis of the shared DEGs.** (A) Top 20 GO Biology process analysis enrichments of the shared 117 DEGs of BM$^{+/−365}$ and BM$^{oim/+}$ ($Q \leq 0.05$). Multiple immune-related DEGs were enriched in cellular response to interferon. The size of the spot represents the number of differential genes, the color represents the Q value; (B, C) Real-time PCR identified the expression levels of *Lfit1* (B) and *Irf7* (C) in BM cells isolated from 4-weeks old wt, heterozygous *Col1a1$^{+/−365}$* and *Col1a2$^{oim/+}$* mice ($n = 3$, $P < 0.05$). The result indicated that both of the two genes were significantly up-regulated in BM$^{+/−365}$ and BM$^{oim/+}$. Data in the quantitative plots are presented as mean ± SD using upaired t-test.

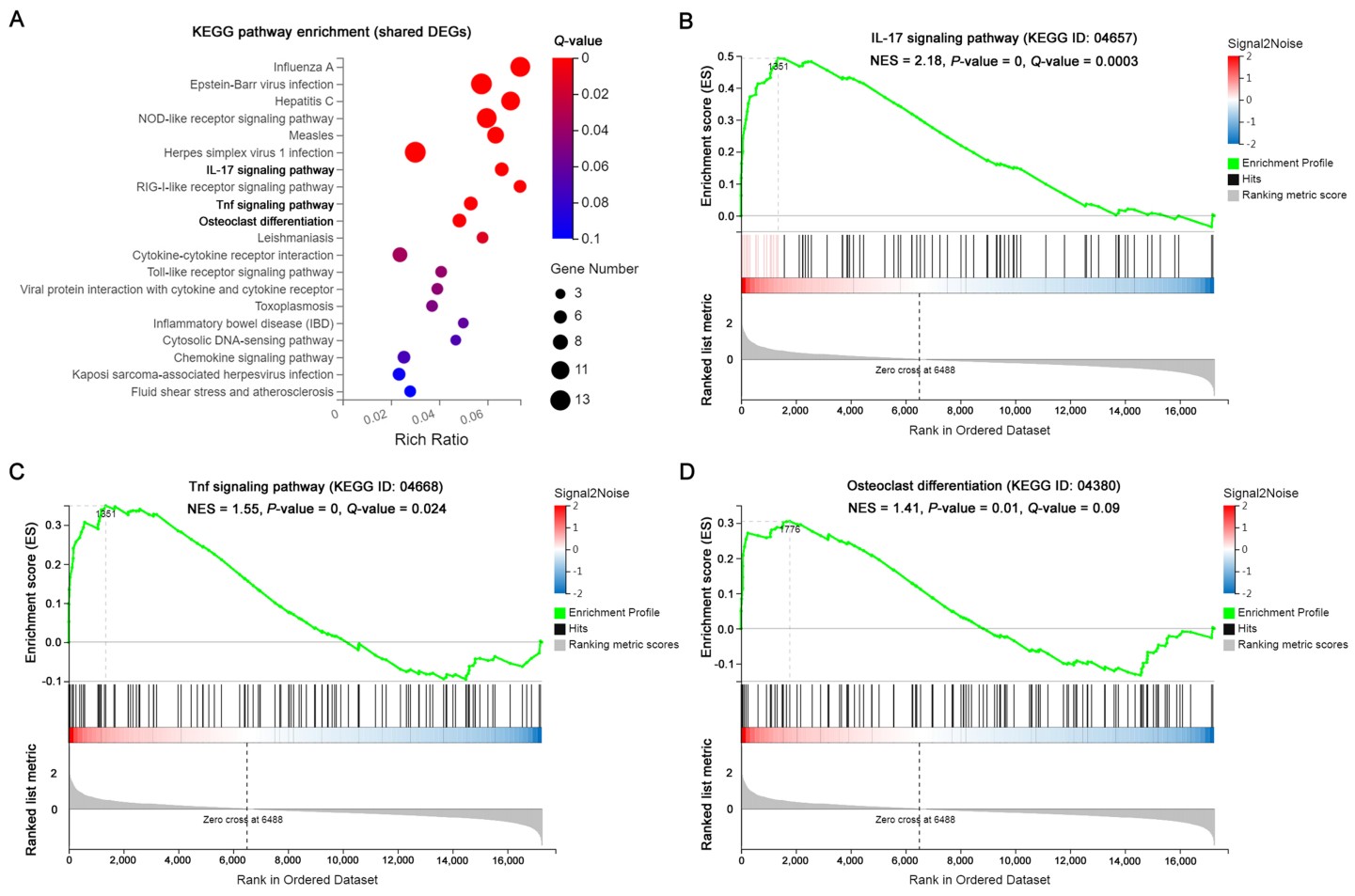

**Figure 3 The results of KEGG pathway enrichment and GSEA analysis of the shared DEGs.** (A) Top 20 KEGG analysis enrichments of the shared 117 DEGs of BM$^{+/-365}$ and BM$^{oim/+}$. The significantly enriched pathways of the shared DEGs included the IL-17 signaling pathway, Tnf signaling pathway and osteoclast differentiation in OI BM cells ($Q \leq 0.05$). The size of the spot represents the number of differential genes, the color represents the Q value. (B–D) The GSEA analysis of the aforementioned three pathways ($P \leq 0.05$ and $Q \leq 0.25$). The results demonstrated that all of the three pathways were markedly upregulated in OI-derived BM cells.

Besides, the expression of these genes in adult mice was also detected by RT-PCR. Data showed that *Tnfα* (Fig. S1C), *Lif* (Fig. S1D), and *Ccl2* (Fig. S1E) were still elevated in mutant BM cells when compared to the normal cells, although there were no differences in the expression of *Ifit1* (Fig. S1A), *Jun* (Fig. 1F), *Stat1* (Fig. S1G) and *Cxcl10* (Fig. S1H). The obviously increased level of *Irf7* (Fig. S1B) and *Ifng* (Fig. S1I) in BM$^{oim/+}$ instead of BM$^{+/-365}$ could be observed in RT-PCR assay. The IL-17 signaling pathway, Tnf signaling pathway and osteoclast differentiation under OI background remained up-regulated until adulthood.

## DISCUSSION

OI is mainly caused by the defective production or assembly of type I collagen, making bone tissue severely affected (*Forlino et al., 2011*; *Hoyer-Kuhn, Netzer & Semler, 2015*). OI patients with *COL1A1* or *COL1A2* mutations can be classified into OI type I-IV

**Table 4 The shared DEGs involved in 'IL-17 signaling pathway', 'Tnf signaling pathway' and 'osteoclast differentiation' of BM$^{+/-365}$ and BM$^{oim/+}$ when compared with BM$^{wt}$.**

| Gene | Description | Log2 (Fold chang) | | Q-value | |
|------|-------------|-------------------|--|---------|--|
| | | BM$^{+/-365}$ | BM$^{oim/+}$ | BM$^{+/-365}$ | BM$^{oim/+}$ |
| Ccl2 | chemokine (C-C motif) ligand 2 | 2.73 | 2.92 | 5.23025917899991e−16 | 0.00000124536325640583 |
| Gm5431 | predicted gene, 45935 | 1.11 | 1.47 | 0.0421644826331762 | 4.85491962362027e−9 |
| Cxcl10 | chemokine (C-X-C motif) ligand 10 | 2.09 | 1.57 | 0.0009539010036805 | 0.0028754859114639 |
| Lif | leukemia inhibitory factor | 1.46 | 1.70 | 0.0098724762681209 | 0.0024271897572989 |
| Ifng | interferon gamma | 1.71 | 2.15 | 0.0002510762900628 | 2.01552889198521e−7 |
| Fosb | FBJ osteosarcoma oncogene B | 2.40 | 2.23 | 0.00000208273094362867 | 0.0008131293868126 |
| Stat1 | signal transducer and activator of transcription 1 | 1.27 | 1.47 | 0.0000710335367185181 | 4.59990999943035e−23 |
| Jund | jun D proto-oncogene | 1.00 | 1.04 | 0.0015850803802578 | 2.41298702546188e−14 |
| Jun | jun proto-oncogene | 1.86 | 1.66 | 4.2489456147999705e−20 | 1.75119110664004e−52 |
| Lfi47 | interferon-induced protein 47 | 1.62 | 1.75 | 0.0247317221652343 | 5.43681272346479e−24 |
| Fcgr1 | Fc receptor, IgG, high affinity I | 2.13 | 2.63 | 0.0033203206673691 | 1.34407141389434e−33 |

**Table 5 Some key genes that were not shared DEGs involved in 'IL-17 signaling pathway', 'Tnf signaling pathway' and 'osteoclast differentiation' of BM$^{+/-365}$ and BM$^{oim/+}$ when compared with BM$^{wt}$.**

| Gene | Description | Log2 (Fold Chang) | | Q-value | |
|------|-------------|-------------------|--|---------|--|
| | | BM$^{+/-365}$ | BM$^{oim/+}$ | BM$^{+/-365}$ | BM$^{oim/+}$ |
| Tnf | tumor necrosis factor | 0.77 | 1.21 | 0.0110345421756999 | 2.8259877145669397e−13 |
| Il17a | interleukin 17A | 2.98 | 4.62 | 0.529929360546415 | 0.0204958961911906 |
| Tnfrsf11a | tumor necrosis factor receptor superfamily, member 11a, NFKB activator/Rank | −0.004 | 0.20 | 0.996564355301463 | 0.791864645628747 |
| Tnfsf11 | tumor necrosis factor (ligand) superfamily, member 11/Rankl | 1.09 | 0.95 | 0.587237549231164 | 0.486382634245118 |

phenotypes (*Forlino et al., 2011*). OI type I due to decreased synthesis of collagen fibrils shows the mildest phenotype, while structurally aberrant type I collagen can give rise to much more severe type II–IV. Despite the subtypes and caused pathogenic variants, high bone turnover represented by hypercellular osteocytes but insufficient mineralization together with enhanced bone absorption generally exist in collagen defect-related OI (*Liu et al., 2019*; *Lopez Franco et al., 2005*; *McBride, Shapiro & Dunn, 1998*; *Saban et al., 1996*). Many studies have focused on the mechanism underlying the unit bone formation insufficiency, while excessive bone erosion has not been extensively explored.

Bone resorption is mainly mediated by osteoclasts that originate from HSC-derived monocytes (*Boyle, Simonet & Lacey, 2003*). The BM monocytes can differentiate into mature multinucleated osteoclasts in the presence of macrophage-colony stimulating factor (M-CSF) and receptor activator of nuclear factor kappa B ligand (RANKL) (*Boyle, Simonet & Lacey, 2003*; *Feng, Guo & Li, 2019*). It has been proved that inflammatory cytokines such as TNFα can act synergistically with RANKL to promote osteoclastotogenesis (*Zhao et al., 2012*). Recently, the inflammatory component in OI pathogenesis has been increasingly attractive. The chronic inflammation state of OI

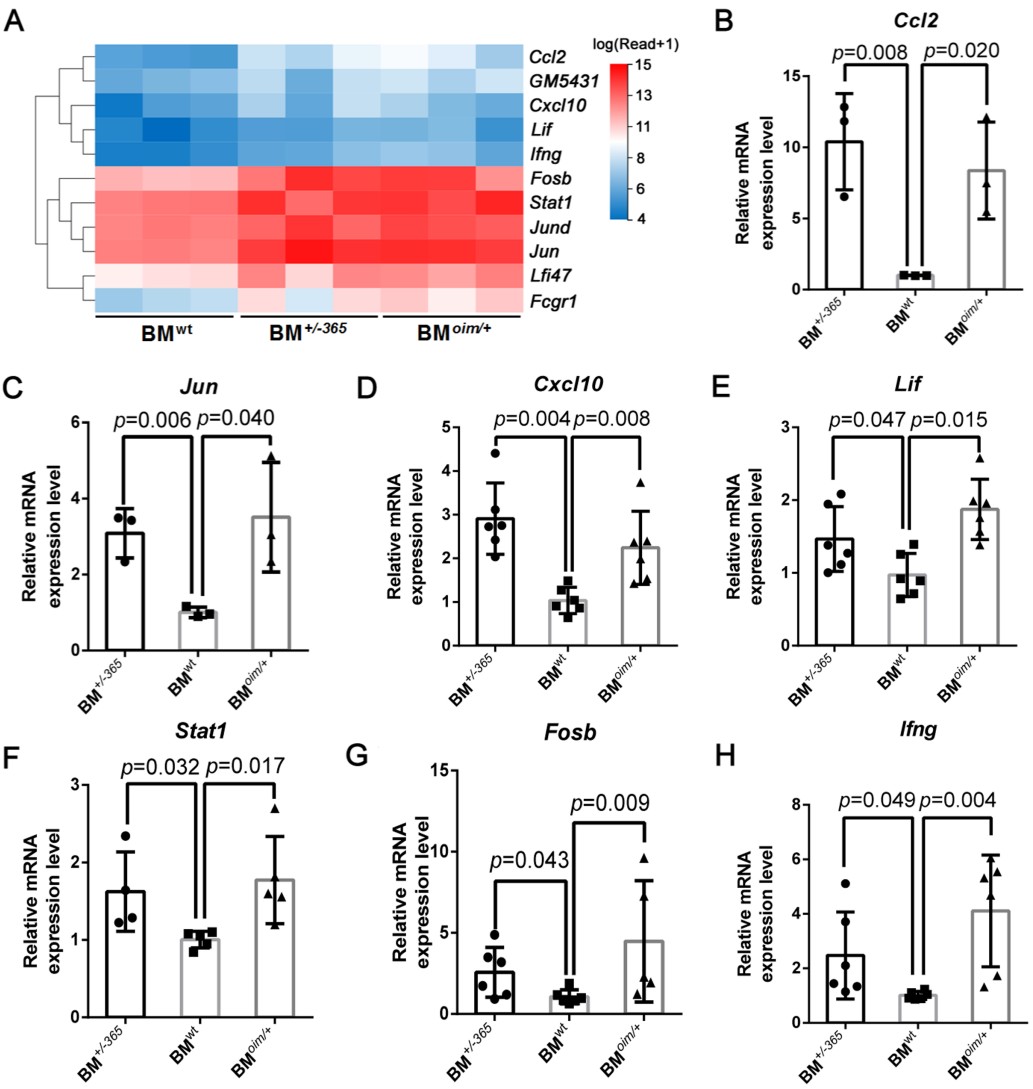

**Figure 4 The expression of the shared DEGs involving IL-17 signaling pathway, Tnf signaling pathway and osteoclast differentiation in BM$^{+/-365}$ and BM$^{oim/+}$ when compared with BM$^{wt}$.** (A) The heatmap of the 11 DEGs involved in IL-17 signaling pathway, Tnf signaling pathway and osteoclast differentiationof 4-weeks BM cells. (B–G) Real-time PCR tested the expression levels of some DEGs. The result indicated that *Ccl2* (B) and *Jun* (C) were apparently upregulated in mutant BM cells. The obviously increased level of *Cxcl10* (D) in BM$^{+/-365}$ together with *Lif* (E) and *Stat1* (F) in BM$^{oim/+}$ could be observed. *Fosb* (G) and *Ifng* (H) also showed an significantly increased in OI BM cells (*n* = 3 for B, C, *n* = 6 for D, E, G, H, *n* = 4 for F, *P* < 0.05). Data in the quantitative plots are presented as mean ± SD using upaired t-test (B–F, H) and Mann-Whitney test (G).

murine models and pediatric patients has been suggested by their elevated serum levels of inflammatory cytokines including IFN and TNFα (*Zhytnik et al., 2020*; *Brunetti et al., 2016*). Many clues suggest that the cells and cytokines of the BM niche can regulate the maturation and function of osteoclasts through direct contact and the paracrine effect (*Herrmann & Jakob, 2019*; *Ciucci et al., 2015*). But the inflammatory changes in BM cells are still unclear.

OI is a type of congenital genetic disease, with the most severe symptoms in childhood (*Paterson, McAllion & Stellman, 1984*). Therefore, exploration of the molecular changes of bone marrow cells of OI in early childhood is helpful to understand its pathogenesis.

Here, we performed RNA-seq of whole BM cells isolated from two types of OI models that respectively stimulate abnormal quantity and structure collagen-related OI. Data showed that the BM cells from the two young murine models shared 117 DEGs ($Q \leq 0.05$, $|Log_2FC| \geq 1$) (Fig. 1). A total of 17 of the top 20 upregulated shared DEGs were the same genes, including several Ifn signaling-related genes containing *Lfit1*, *Lfit3*, *Lfi44* and *Irf7* (Tables 2 and 3). Consistent with the findings of Zhytnik's group, GO biology process analysis also enriched multiple DEGs involved in the cellular response to Ifnβ and Ifnα in defective BM cells (Fig. 2), suggesting the activated Ifn signaling in OI BM. Transcription factor STAT1 is a key effector of the IFN pathway (*Michalska et al., 2018*). The upregulated *Stat1* in OI BM cells (Fig. 4F) further revealed the dysregulated Ifn signaling. IFNs are key cytokines for both innate and adaptive immune responses (*Takayanagi et al., 2005*). Some previous studies found that IFN-γ can promote osteoclastogenesis under T-cells activation and enhance the multinucleation of myeloid lineage cells in osteoporosis (*Biros et al., 2022*). Thus the overactivated Ifn signaling might also act on the bone resorption in the OI background.

KEGG and GSEA analysis of the 117 shared DEGs showed that the IL-17 signaling pathway, Tnf signaling pathway and osteoclast differentiation were significantly upregulated in OI BM cells (Fig. 3). A total of 11 enriched DEGs were involved in the three pathways and displayed complex crosstalk (Table 4). *Il17a*, *Tnf* and *Rankl* showed different degrees of upregulation (Table 5), indicating the chronic inflammation and enhanced osteoclastogenesis. RT-PCR assay confirmed the elevated expression of some DEGs in young and adult mice (Fig. 4, Fig. S1). The results of 4-weeks old mice were consistent with the results of RNA-seq (Fig. 4). While only several genes remained elevated in 12-weeks old OI mice (*Tnfα*, *Lif*, and *Ccl2*) (Fig. S1), which might be related to the relieved bone phenotype in adulthood. IL-17A and TNFα have been proved to play an essential role in inflammatory bone erosion by inducing the production of RANKL (*Weitzmann, 2017*; *Zhao et al., 2012*). IL-17A can also promote TNFα secretion. *Matthews et al. (2017)* found the increased serum level of Tnf in homozygous oim mice, however, anti-TNFα therapy failed to reduce bone absorption. These results suggested that blocking Tnf signaling alone is insufficient to effectively reverse the excessive bone resorption in OI bones. Targeting both IL-17 and Tnf signalings may be an efficient strategy for OI treatment.

## CONCLUSIONS

This study preliminarily revealed the dysregulated biological process of cellular response to Ifn together with upregulated IL-17 signaling, Tnf signaling and osteoclast differentiation in young male OI BM niche by RNA-seq. Either defective collagen production or abnormal collagen assembly shared similar alterations in gene profiles and pathways involving inflammation and osteoclast activation. Data presented here not only contributed to understanding the mechanism of the enhanced bone absorption in the bone of OI, but also provided more evidence to develop potential anti-inflammation therapies.

### Funding

The present study was supported by grants from the National Key R & D Program of China (2017YFC1001904). The funders had no role in study design, data collection and analysis, decision to publish, or preparation of the manuscript.

### Grant Disclosures

The following grant information was disclosed by the authors:
National Key R & D Program of China: 2017YFC1001904.

### Competing Interests

The authors declare that they have no competing interests.

### Author Contributions

- Chenyi Shao conceived and designed the experiments, performed the experiments, analyzed the data, prepared figures and/or tables, authored or reviewed drafts of the article, and approved the final draft.
- Yi Liu conceived and designed the experiments, analyzed the data, prepared figures and/or tables, authored or reviewed drafts of the article, and approved the final draft.
- Jiaci Li conceived and designed the experiments, performed the experiments, prepared figures and/or tables, authored or reviewed drafts of the article, and approved the final draft.
- Ziyun Liu performed the experiments, prepared figures and/or tables, authored or reviewed drafts of the article, and approved the final draft.
- Yuxia Zhao analyzed the data, authored or reviewed drafts of the article, and approved the final draft.
- Yaqing Jing analyzed the data, authored or reviewed drafts of the article, and approved the final draft.
- Zhe Lv analyzed the data, authored or reviewed drafts of the article, and approved the final draft.
- Ting Fu analyzed the data, authored or reviewed drafts of the article, and approved the final draft.
- Zihan Wang analyzed the data, authored or reviewed drafts of the article, and approved the final draft.
- Guang Li analyzed the data, authored or reviewed drafts of the article, and approved the final draft.

### Animal Ethics

The following information was supplied relating to ethical approvals (*i.e.*, approving body and any reference numbers):

The study was approved by the Animal Care and Use Committee of Tianjin Medical University (TMUaMEC 2017012).

## Ethics

The following information was supplied relating to ethical approvals (*i.e.*, approving body and any reference numbers):

The study was approved by The Animal Ethical and Welfare Committee (AEWC) (TMUaMEC 2017012).

## Data Availability

The RNA-seq data is available at NCBI SRA: PRJNA835622.

## Supplemental Information

Supplemental information for this article can be found online at http://dx.doi.org/10.7717/peerj.13963#supplemental-information.

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
