# Peer review of "Up-regulated IL-17 and Tnf signaling in bone marrow cells of young male osteogenesis imperfecta mice"

_PeerJ, doi:10.7717/peerj.13963_

## Round 0.1 · original submission · Major Revisions

Please, respond to the Reviewers' questions/comments and make sure that the correct statistics are used. Also, a sample of older mice (or available data from literature) should be used as both Reviewers point out. Using only 4 weeks old mice may not be enough to draw conclusions on adults (including middle-aged and or old mice).

Reviewer 1 ·

Basic reporting

'no comment'

Experimental design

'no comment'

Validity of the findings

'no comment'

Additional comments

In this work by Chenyi Shao et al the authors have addressed a novel aspect of OI in the
context of inflammation.

I am interested to know why the authors used 4 weeks old mice bone marrow instead of
adult bone marrow.

Does the authors expect IL-17 and TNF related gene changes in adult BM similar to young
mice? It would be interesting to do a qpcr of the genes in adult mouse BM and compare
to the 4 week BM.

Annotated reviews are not available for download in order to protect the identity of reviewers who chose to remain anonymous.

Reviewer 2 ·

Basic reporting

N/a

Experimental design

N/a

Validity of the findings

N/a

Additional comments

I do have some major concerns regarding the data presented by the authors.

The authors have only used male mice for the study. However, there are several studies showing dramatic sex dependence on bone structure and morphological properties in OI mouse models. These features could lead to changes in the expression various differentially expressed genes or vice versa. The authors have not provided any justification why only this specific sex was studied. The title of the manuscript should include male OI mice with a detailed description in the methods or results section.

RT-PCR validation here shown by the authors were not consistent with sequencing data which authors have also stated (pg 10, line 169-172). The authors further state that the different samples used in RNA-seq and RT-PCR assay might be the main reason for this inconsistency. This questions the reliability of the DEG results. Why different samples were used when authors should have used the same sample/mouse for both the analyses. Can one use tibia and femur both for such analysis? Please clarify. Again, the low sample size could also be the reason for such inconsistencies.

The authors have only analyzed mice at 4 weeks of age. I strongly believe that the authors should have done a comparison between young and old mice as the DEG's expression may vary at different time points. If this is never been tested then please explain why only young mice were used for this study?

Minor concerns:

1) Please read the manuscript thoroughly there are some errors and typos for eg. in results section for fig 4 description everywhere its referred as fig 2.

2) Figure legends should be more descriptive.

Statistical concerns:

The authors are not consistent with the p values presentation on the bar graphs. Some of the graphs shows a p value and others do not.

For gene expression analysis what is the method used to interpret the data? The sample size used for the gene expression analysis is low with high SD values.
In general, the figure legends should include details on n-number in each experiment and statistics.

---

## Round 0.2 · accepted · Accept

The Reviewer felt that you responded to all of his comments, suggestions and is satisfied with the revised manuscript.

Reviewer 2 ·

Basic reporting

na

Experimental design

na

Validity of the findings

na